# GinSign: Grounding Natural Language Into System Signatures for Temporal Logic Translation

## Abstract

Natural language (NL) to temporal logic (TL) translation enables engineers to specify, verify, and enforce system behaviors without manually crafting formal specifications—an essential capability for building trustworthy autonomous systems. While existing NL–to-TL translation frameworks have demonstrated encouraging initial results, these systems either explicitly assume access to accurate atom grounding or suffer from low grounded translation accuracy. In this paper, we propose a framework for **G**rounding Natural Language **In**to System **Sign**atures for Temporal Logic translation called GinSign. The framework introduces a grounding model that learns the *abstract task* of mapping NL spans onto a given system signature: given a lifted NL specification and a system signature $\mathcal{S}$, the classifier must assign each lifted atomic proposition to an element of the set of signature-defined atoms $\mathcal{P}$. We decompose the grounding task hierarchically—first predicting *predicate* labels, then selecting the appropriately typed *constant* arguments. Decomposing this task from a free-form generation problem into a structured classification problem permits the use of smaller masked language models and eliminates the reliance on expensive LLMs. Experiments across multiple domains show that frameworks which omit grounding tend to produce syntactically correct lifted LTL that is semantically nonequivalent to grounded target expressions, whereas our framework supports downstream model checking and achieves grounded logical-equivalence scores of $95.5\%$, a $1.4\times$ improvement over SOTA.

## 1 Introduction

Formal language specifications are foundational to a wide range of systems, including autonomous robots and vehicles (Tellex et al., 2020; Raman et al., 2013; Mallozzi et al., 2019; Harapanahalli et al., 2019), cyber-physical controllers (Konur, 2013; Abbas et al., 2013; Hoxha et al., 2018), and safety-critical software (Alur, 2015; Yoo et al., 2009; Bowen & Stavridou, 1993). Among these, temporal logic (TL) specification plays a central role in enabling formal verification of these systems (Watson & Scheidt, 2005; Bellini et al., 2000). Despite its power, TL specification is difficult and typically requires domain expertise (Yin et al., 2024; Cardoso et al., 2021; Thistle & Wonham, 1986). In practice, system requirements are often provided by stakeholders in natural language (NL), which is inherently ambiguous and lacks formal precision (Veizaga et al., 2021; Lamar, 2009; Lafi et al., 2021). To bridge this gap, there has been growing interest in using artificial intelligence methods to automatically translate natural language specifications into temporal logic (Fuggitti & Chakraborti, 2023; Chen et al., 2023).

Despite recent gains from deploying sequence-to-sequence (seq2seq) (Hahn et al., 2022; Pan et al., 2023; Hsiung et al., 2022) and large language models (LLMs) (Fuggitti & Chakraborti, 2023; Chen et al., 2023; Xu et al., 2024; Cosler et al., 2023), most NL-to-TL translation pipelines remain nondeployable in real-world systems. While lifted translation systems often yield well-formed and valid TL specifications, the atomic propositions (APs) over which the TL operates are never explicitly defined. This is a deliberate choice by SOTA methods which explicitly assume accurate groundings are readily available and the mappings are known (Hsiung et al., 2021; Chen et al., 2023). However, without a semantic definition for each AP, which is grounded in a system and the world in which the system operates, the resulting TL formulas cannot be interpreted on traces or state machines, making

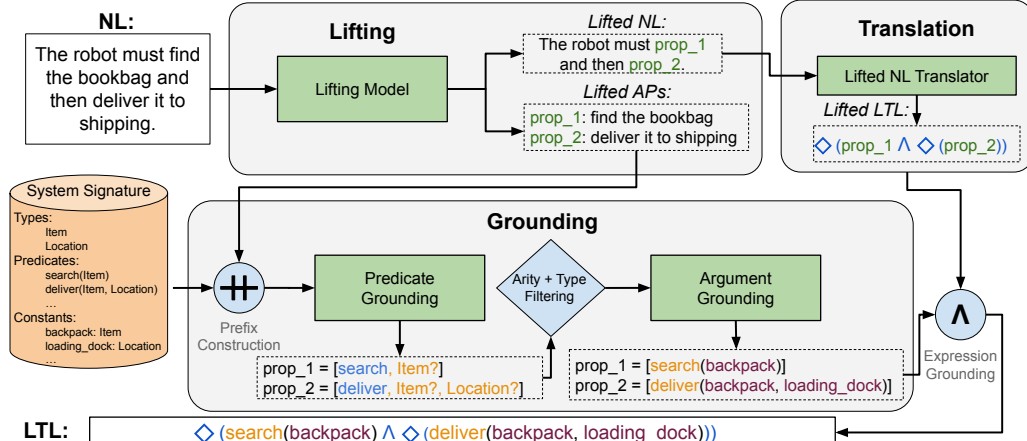

Figure 1: Overview of the GinSign Framework.

real-world deployment impossible. While there has been significant progress on *visual grounding* (linking natural language expressions to entities in perceptual scenes), these methods target perceptual reference resolution rather than the formal symbolic grounding considered in this work (Qiao et al., 2020). In our setting, Lang2LTL (Liu et al., 2023) employs an embedding-similarity approach to align lifted APs with semantic maps. However, this method undershoots state-of-the-art lifted translation accuracy by more than 20%, highlighting the challenge of achieving both accurate lifting and precise grounding simultaneously.

To illustrate this challenge, consider the NL translation shown in the top half of Figure 1. Many systems translate, "The robot must find the bookbag and then deliver it to shipping.", to $\diamondsuit(prop_1 \wedge \diamondsuit(prop_2))$, which is logically coherent but semantically useless. Because the system's underlying predicates ($search$, $deliver$) and constants ($backpack$, $loading\_dock$) are not incorporated, the translation lacks operational meaning. This disconnect between syntax and semantics is a primary bottleneck for deploying NL-to-TL systems in grounded, dynamic environments. To address this gap, we posit that effective NL-to-TL translation for real-world deployment must ground all APs, which is shown in the bottom of Figure 1.

In this paper, we propose GinSign, a framework for **G**rounding Natural Language **In**to System **Sign**atures for Temporal Logic. Our framework bridges the current disconnect between theoretical and practical application by both (1) inferring logical structure (temporal operators, boolean connectives), and (2) grounding NL spans to an inventory of atomic propositions with clear semantic grounding within a system. Our main contributions are as follows:

- We generalize the AP grounding task into a multi-step classification problem: given an NL string (lifted AP) and a system signature (potentially containing unseen classes), classify the NL span into a predicate and its arguments, grounded in the system signature.

- We propose a solution to the above problem using a hierarchical grounding framework. First, intermediate grounding is performed to classify the AP into a predicate on the system signature via an encoder-only model. Then, using prior knowledge of the predicate arity and types defined in the system signature, the same model is used to ground the typed arguments of that predicate.

- In experimental evaluation, GinSign outperforms LLM baselines in the grounded translation task by up to $1.4\times$. We establish a new baseline for end-to-end grounded translation on multiple datasets, presenting NL-to-TL translation that is executable on real traces.

The remainder of this paper is organized as follows. Section 2 formalizes NL-to-TL translation. Related work is surveyed in Section 3. Section 4 presents our grounding-first framework. Section 5 reports empirical results and analyzes common failure modes. Section 6 concludes this work with a discussion of limitations and directions for future work.

## 2 BACKGROUND

In this section, we provide preliminaries on linear temporal logic, we motivate the use of system signatures in the natural language grounding problem, we formulate the NL-to-TL translation task, and we discuss current SOTA NL-to-TL translation approaches.

### 2.1 LINEAR TEMPORAL LOGIC

The syntax of LTL is given by the following grammar:

$$\varphi ::= \pi \mid \neg\varphi \mid \varphi_1 \wedge \varphi_2 \mid \varphi_1 \vee \varphi_2 \mid \varphi_1 \Rightarrow \varphi_2$$
$$\mid \bigcirc\varphi \mid \Diamond\varphi \mid \Box\varphi \mid \varphi_1 \, \mathcal{U} \, \varphi_2$$

Where each atomic proposition belongs to a set of known symbols $\pi \in \mathcal{P}$. To verify a real system against an LTL requirement, one typically models the implementation as a finite-state Kripke structure $\mathcal{M} = (S, S_0, R, L)$, where $S$ is the state set, $S_0 \subseteq S$ the initial states, $R \subseteq S \times S$ the total transition relation, and $L : S \to 2^{\mathcal{P}}$ the labeling function. Because every atomic proposition in $\varphi$ is interpreted via $L$, *grounding* those propositions in the signature of $\mathcal{M}$ is prerequisite to even running the model checker. In other words, the syntactic formula only becomes semantically testable once its APs are linked to concrete predicates over system states (Hsiung et al., 2021).

### 2.2 SYSTEM SIGNATURES

Well-designed automated systems and planners are typically grounded in a *planning domain definition language* (Ghallab et al., 1998), action vocabulary, or some other domain-specific semantic library for a system (Zhang et al., 2024; Oswald et al., 2024). We observe that these languages are realizations of many-sorted logical systems, and are therefore motivated to further apply this formalism in our efforts to ground TL specifications. Accordingly, we look to system signatures as the formal vocabulary that ties a grounded TL specification to the structure it specifies, as any well-formed system should have a system signature. Formally, a many-sorted system signature is defined as follows:

$$\mathcal{S} = \langle T, P, C \rangle$$

where $T$ is a set of *type* symbols, $P$ is a set of *predicate* symbols, and $C$ is a set of *constant* symbols. System signatures are used to describe all of the non-logical terminals in a formal language (Finkbeiner & Zarba, 2006).

Each component of $\mathcal{S}$ plays a distinct but interconnected role. Types $t \in T$ act as categories that restrict how constants and predicates may be combined—for example, distinguishing between arguments of type `location`, `agent`, or `item`. Constants $c \in C$ are the specific instances of these types, such as a particular `location` like `loading_dock`, or a particular `item` like `apple`. Predicates $p \in P$ then specify relations or properties defined over typed arguments: $p(t_1, \ldots, t_m)$ requires arguments drawn from the corresponding type sets, yielding well-typed atoms like `deliver(apple, loading_dock)`. Thus, the connection between types, constants, and predicates informs the structure of possible grounding targets: constants instantiate types, and predicates bind these constants together into statements about the world.

### 2.3 NL-TO-TL TRANSLATION

In this section, we review the natural language to temporal logic (NL-to-TL) translation task. Prior work divides the task into the following three phases: lifting, translation, and grounding. To make this process concrete, we illustrate each step with the example specification: "`Eventually pick up the package from room A.`"

**Lifting:** We define *lifting* as the following classification problem. Given a sequence of tokens that constitute an NL specification $\mathbb{S}$, perform integer classification on each token as either a reference to some $\pi$, or not. Formally:

$$\lambda : \mathbb{S} \to \begin{cases} 0, & \mathbb{S}_i \textbf{ is not} \text{ part of any } \pi \in \mathcal{P}, \\ n, & \mathbb{S}_i \textbf{ is} \text{ part of } \pi_n. \end{cases}$$

The result of successful lifting is a mapping $\lambda \colon \mathbb{S} \to \{i \mid i \in \mathbb{Z}\}$ of lifted substrings to integer AP references that appear in the corresponding LTL expression. In LTL, each atomic proposition $\pi$ is a boolean variable whose value is determined at evaluation time. The value is given by the presence of that AP on a trace. *Example.* In the sentence above, the phrase "`pick up the package from room A`" is identified as a reference to an atomic propositions. Lifting replaces them with placeholders, producing: "`Eventually prop`$_1$`.`"

**Translation:** Given a natural-language specification $s$, the goal is to produce a temporal-logic formula $\varphi$ that preserves the intended behavior: $f \colon s \longrightarrow \varphi$. Let $\mathcal{P} = \{\pi_1, \ldots, \pi_m\}$ be the finite set of atomic propositions, each with a semantic interpretation over traces (e.g., $[\![\pi_i]\!] \subseteq \Sigma^\omega$). A TL formula is built from $\mathcal{P}$ using boolean and temporal operators (e.g., $\diamondsuit, \square, \bigcirc, \mathcal{U}$). For $f(s)$ to be actionable, every $\pi_i$ appearing in $\varphi$ must be mapped to a meaning defined in the system signature $\mathcal{S}$, outlined in section 2.2. *Example.* The lifted string "`Eventually prop`$_1$" is translated into the LTL formula $\varphi = \diamondsuit\, \texttt{prop}_1$.

**Grounding:** Let the lifted specification contain $k$ placeholder atoms $\{\texttt{prop}_1, \ldots, \texttt{prop}_k\}$. Grounding is defined as a total function

$$g_\mathcal{S} \colon \{\texttt{prop}_1, \ldots, \texttt{prop}_k\} \longrightarrow \underbrace{\big\{ p \mid p \in P \big\} \,\cup\, \big\{ p(c_1, \ldots, c_m) \mid p \in P,\ c_i \in C \big\}}_{\mathcal{P}_\mathcal{S}},$$

which assigns to every placeholder either (i) a predicate $p \in P$ (nullary case) or (ii) a fully instantiated atom $p(c_1, \ldots, c_m)$ whose arguments $c_i$ are constants of the appropriate type. The image set $\mathcal{P}_\mathcal{S}$ thus represents the *grounded* atomic-proposition vocabulary permitted by $\mathcal{S}$. For the remainder of the paper, we will refer to constants as *arguments*, as they are used exclusively as arguments accepted by predicates in $P$. *Example.* Given the system signature $\mathcal{S} = \langle T = \{\texttt{room}, \texttt{object}\}, P = \{\texttt{pick\_up(obj, room)}\}, C = \{\texttt{roomA}, \texttt{package1}\}\rangle$, the grounding step resolves $\texttt{prop}_1 \mapsto \texttt{pick\_up(package1, roomA)}$. The final grounded formula is: $\varphi = \diamondsuit\, \texttt{pick\_up(package1, roomA)}$.

## 3 RELATED WORK

Current frameworks for neural NL-to-TL translation either attempt translation in one shot using an LLM prompting approach, or divide the task into multiple steps, including lifting, verification, and human-in-the-loop feedback. An overview is provided in Table 1.

**NL2LTL:** (Fuggitti & Chakraborti, 2023) introduce a Python package that implements a few-shot, template-based approach to NL-to-LTL translation. Users of this package are required to supply LTL templates and example translations in order to construct the prompt.

**NL2TL:** (Chen et al., 2023) introduce a framework that decomposes the end-to-end translation task into 1) lifting with a GPT model and 2) translation with a seq2seq model. Most critically, this approach reports that lifting is an effective method for reducing translation complexity. We continue to exploit this fact in our framework, yet, we find this lifted translation can not be *verified* until the lifted APs have been grounded—a step omitted from this framework, as shown in the table.

**Lang2LTL:** (Liu et al., 2023) propose a modular pipeline for language grounding and translation into LTL. Their framework separates the task into referring expression recognition with an LLM, grounding through embedding similarity against a semantic database. Yet it presumes access to proposition embeddings from a semantic map and does not extend this mechanism to arbitrary system signatures, which we perform in our work.

| Framework | Lifting | Translation | Grounding |
|---|---|---|---|
| LLM-Baseline | ✗ | ✓ | ✗ |
| NL2LTL (Fuggitti & Chakraborti, 2023) | ✗ | ✓ | ✗ |
| NL2TL (Chen et al., 2023) | ✓ | ✓ | ✗ |
| Lang2LTL (Liu et al., 2023) | ✓ | ✓ | ✓ |
| GinSign (ours) | ✓ | ✓ | ✓ |

Table 1: Overview of each framework's support for lifting, grounding, and translation.

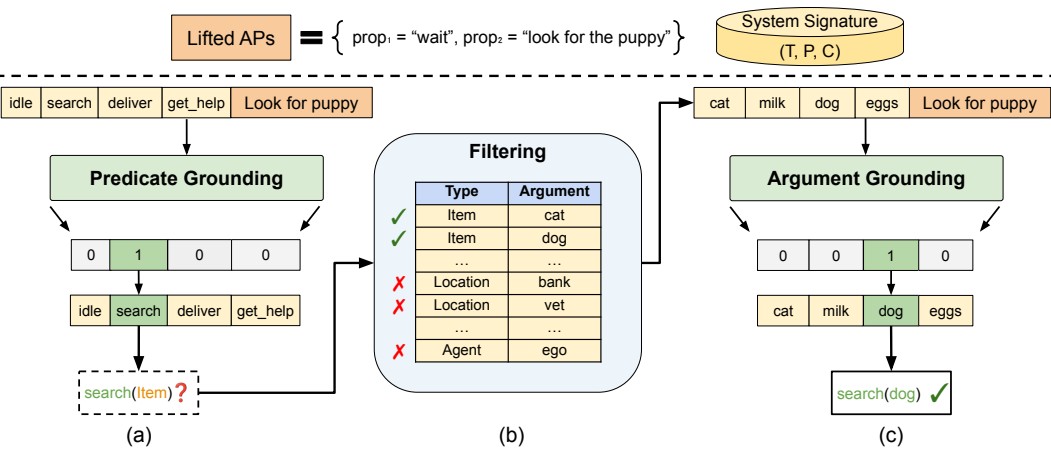

Figure 2: An overview of our grounding components. Given $n$ lifted NP APs, we convert the system signature into a prefix using Algorithm 1. The lifted NL is first combined with the predicates prefix to ground the predicate to a known action (a). Since each predicate requires an argument, we filter out non-candidate arguments by type (b). We then combine the lifted NL with the arguments prefix to classify the correct argument (c). Both predicate and argument grounding use the same token classification BERT model, which processes any prefix and lifted NL.

# 4 METHODOLOGY

In this section, we introduce GinSign, an end-to-end *grounded* NL-to-TL translation framework. GinSign accepts two input components: a system signature, and a natural language specification. We leverage the compositional approach and decompose the NL to TL translation task into lifting, translation, and grounding. We perform the lifting and translation using a BERT (Devlin et al., 2019) and T5 (Raffel et al., 2020) model as in previous works. Our methodology focuses on the grounding of the APs obtained from the lifting into the defined state space. The grounded APs will then be inserted into the temporal logic formula obtained from the lifted translation.

The input to the grounding module is the NL string associated with each extracted AP. The output is the grounding of this NL string into the state space defined using a system signature. The GinSign framework performs the grounding using a hierarchical approach consisting of predicate grounding and argument grounding. An overview of the framework is shown in Figure 2 and the details are provided in Section 4.1. Both of the grounding steps are formulated as an *abstract grounding task*, which we solve using a BERT model. The details are provided in Section 4.2.

## 4.1 HIERARCHICAL GROUNDING

In this section, we first describe the hierarchical grounding strategy of GinSign with Figure 2. Next, we provide the details of the predicate grounding and the argument grounding.

**Hierarchical strategy:** We design a hierarchical grounding strategy because of the dependency between the predicates and arguments. Each predicate has a fixed number of arguments with specified type. The hierarchical decomposition can by design ensure that the correct number of arguments of the right type are assigned to each predicate. The two grounding steps are defined, as follows:

1. **Predicate grounding**: predicts a predicate $p \in P$ for each placeholder, reducing the argument search space to $C_t = \{ c \in C \mid \text{type}(c) = t \}$ (Figure 2 (a)).

2. **Argument grounding**: dependent on $p$, chooses concrete constant(s) from $C_t$ and produces the final atom $p(c_1, \ldots, c_m)$, thereby completing $g_S$ (Figure 2 (c)).

This hierarchy turns a single, large open-set decision into two smaller problems. A flat classifier would implicitly consider on the order of $\sum_{p \in P} \prod_{r=1}^{\text{arity}(p)} |C_{\tau_r}|$ labelings per instance (all fully-instantiated atoms), which quickly becomes intractable as $|C|$ grows. In contrast, the first stage selects among $|P|$ predicates; the second stage then solves at most $m = \text{arity}(p)$ independent choices, each over

$|C_{\tau_r}|$ typed constants. Concretely, the effective label budget per instance drops from $\Theta\big(\prod_r |C_{\tau_r}|\big)$ to $\Theta\big(|P| + \sum_r |C_{\tau_r}|\big)$, a dramatic reduction when $|C| \gg |P|$ and types partition $C$ evenly. Moreover, type filtering eliminates invalid atoms by construction, ensuring any predicted $p(c_1, \ldots, c_m)$ lies in $\mathcal{P}_\mathcal{S}$. Coupled with windowed classification (Sec. 4.2), each sub-decision operates over at most $m$ candidates at a time, further improving sample efficiency and calibration while preserving exact compatibility with the evaluation-time tournament.

**Predicate Grounding:** In Section 2, we framed grounding as an NL classification task. To ground an input specification, the classifier must treat every symbol in the signature as a potential label. Rather than allocating a fixed soft-max head with one output neuron *per* symbol—as typical token-classification pipelines would—we prepend a rigid, pseudo-natural prefix that *enumerates* the target signature and let the encoder attend over it. The BERT backbone, therefore, learns the abstract skill of scoring span–prefix alignments instead of memorizing a static label inventory. This process is seen in Figure 2 (a) where the input is lifted APs and the system signature predicates. Crucially, the class set is no longer baked into the model parameters: supplying a different prefix instantly defines a new label universe, so the same fine-tuned weights can be reused across domains with disjoint signatures. We later show (in Table 5) that this design yields promising out-of-distribution accuracy—comparable to in-domain performance—by simply swapping prefixes, demonstrating that the model has internalized grounding as a transferable reasoning operation rather than rote classification.

**Filtering:** After predicate grounding, we know which predicate(s) are present in the segment. Using this information, we query the system signature for the arity of the predicate (i.e., the type and number of required arguments). This knowledge filters the search space for argument grounding to the subset of arguments compatible with the predicted predicate's types. We show the transition of information between predicate and argument grounding in Figure 2 (b), where predicate information from the system signature is used to map lifted APs to the known possible arguments.

**Argument Grounding:** Argument grounding is then framed as the further classification of typed natural language spans into specific domain arguments. Each argument is resolved independently against its type-filtered candidate set $L_c^{(r)}$ by the same BERT backbone as before, simply with a different prefix, as seen in Figure 2 (c). Thus, the final output always contains the correct number of arguments, with no need for an additional constraint or stopping criterion.

## 4.2 Grounding Model

We operationalize $g_\mathcal{S}$ as a classifier over an input-defined label set. For each lifted placeholder $\mathrm{prop}_i$, the model receives (i) the local NL context corresponding to the lifted AP and (ii) a *prefix* that enumerates the relevant candidates from $\mathcal{S} = \langle T, P, C \rangle$ up to length maximum input length $m$. The model points to one element of the prefix; because the prefix is constructed from $\mathcal{S}$ at input time, the label universe is domain-agnostic and requires no change to the classifier head across domains.

**Prefix construction.** Let $\mathrm{enum}(\cdot)$ produce a fixed-order list of symbols as a token sequence. For predicate grounding, the candidate list is $L_p = \mathrm{enum}(P)$. For argument grounding, once a predicate $\hat{p} \in P$ with arity $a$ and type signature $(\tau_1, \ldots, \tau_a) \in T^a$ is predicted, the $r$-th argument uses the type-filtered set,

$$L_c^{(r)} \;=\; \mathrm{enum}\big(\{\, c \in C \mid \mathrm{type}(c) = \tau_r \,\}\big).$$

We serialize the input as a pair $(x_{\mathrm{AP}}, x_{\mathrm{prefix}})$, where $x_{\mathrm{prefix}}$ is the tokenization of $L$ (either $L_p$ or $L_c^{(r)}$). Our exact implementation of this process is given in Appendix A.3, in Algorithm 1. When $N > m$, we perform prefix sharding and apply a two-stage *tournament* procedure. First, the model classifies within each shard $W_j$. The winning candidate from each shard is then re-assembled into a new prefix list, and the procedure repeats until a single element remains. This ensures scalability to arbitrarily long prefixes while keeping each classification head fixed in size.

**Classification.** Let $L = [\ell_1, \ldots, \ell_N]$ be the candidate list for the current decision. We fix a shard size $m$ and partition $L$ into contiguous windows

$$W_j \;=\; [\,\ell_{(j-1)m+1}, \ldots, \ell_{\min(jm,N)}\,], \qquad j = 1, \ldots, \lceil N/m \rceil.$$

The model $h_\theta$ maps a pair and a window to a discrete index in $\{1, \ldots, |W_j|\}$:

$$h_\theta : \; (x_{\mathrm{AP}}, W_j) \;\mapsto\; \hat{y} \in \{1, \ldots, |W_j|\}.$$

For short lists ($N \leq m$), we pad $L$ to length $m$ with a `<pad>` token and classify once.

**Training objective.** For each training instance with gold label $\ell^\star \in L$, we construct one or more *gold-in* shards $W_j$ such that $\ell^\star \in W_j$ (contiguous windows). The supervision is a single-label CrossEntropy over the shard positions:

$$\mathcal{L}_{\mathrm{CE}}(\theta) \;=\; -\log\ p_\theta\big(y = \mathrm{index}(\ell^\star \in W_j) \,\big|\, x_{\mathrm{AP}}, W_j\big).$$

Further implementation details can be found in Appendix A.7. Relevant code is available in the supplementary material and will be made publicly available upon acceptance.

## 5 EXPERIMENTS

In this section, we present the results of our experiments and evaluations of our grounding framework, as well as its impact on end-to-end translation. This section is organized as follows. Subsection 5.1 discusses training and evaluation corpora information. The results of our isolated grounding evaluation are presented in Subsection 5.2. Finally, end-to-end translation results are presented in Subsection 5.3.

### 5.1 DATASETS AND METRICS

Table 2: Overview of the magnitude of each domain signature, by field.

| **Domain** ($\mathcal{S}$) | **Types** $|T|$ | **Predicates** $|P|$ | **Arguments** $|C|$ |
|---|---|---|---|
| Search and Rescue | 2 | 7 | 44 |
| Traffic Light | 5 | 4 | 175 |
| Warehouse | 2 | 5 | 82 |

**Datasets** We use VLTL-Bench (English et al., 2025b) as the primary dataset for evaluation, as it is the only resource we found that grounds natural language specifications in a concrete state space. VLTL-Bench consists of three distinct domains (Search and Rescue, Traffic Light, and Warehouse), each providing lifted natural language specifications, grounded LTL formulas, and reference traces. Table 2 provides the magnitude of each element of the system signatures used in these datasets. We note here that while the Traffic Light domain has the greatest number of arguments, the Warehouse domain poses a distinct challenge to argument grounding: constants in this domain are not lexically consistent with their surface realizations in text, a difficulty reflected in our results. Because the grounding task—and particularly the grounded logical equivalence metric—requires access to both lifted and grounded APs, prior datasets such as Navigation (Wang et al., 2021), Cleanup World (MacGlashan et al., 2015), and GLTL (Gopalan et al., 2018) are not applicable and are therefore omitted from our evaluations.

**Metrics** For each evaluation, we report the mean, variance, and confidence of each metric. We compute the following metrics for our evaluations: **LE** (Logical Equivalence, and **GLE** (*Grounded Logical Equivalence*). For the Grounding tasks, we report $F_1$ scores over all APs. For the End-to-End Translation, **LE** is distinguished from **GLE** in that the former does not account for AP grounding in the Linear Temporal Logic, resulting in high scores for an expression such as "$\mathrm{prop}_1 \to \Diamond(\mathrm{prop}_2)$", while *grounded* logical equivalence demands that $\mathrm{prop}_1$ and $\mathrm{prop}_2$ are properly defined in order to be scored as correct. To evaluate grounded logical equivalence, grounded TL is parsed by the pyModelChecking framework (Casagrande, 2024), which will extract the APs, allowing for comparison against the ground truth grounding.

### 5.2 GROUNDING EVALUATION

Here, we discuss the results of the isolated grounding evaluations. We perform three experiments to evaluate the performance of our proposed method against in-context LLM prompting baselines (provided in supplementary materials). First, we report the $F_1$ score achieved by each approach on the predicate and constant grounding task. In the predicate grounding task, each approach receives a lifted

natural language AP to be classified into one or more domain-specific predicates. In the argument grounding task, each approach receives a lifted natural language AP and all constant arguments of the appropriate type, with the goal of grounding into a specific domain constants.

Table 3: Evaluation of grounding approaches. We report $F_1$ (per AP) for both predicate and argument grounding. [†] Note that this framework does not distinguish between predicate and argument grounding. We therefore evaluate overall AP grounding in the **Argument Grounding** column. The prompt used by both GPT models is given in Appendix A.6.

| | Predicate Grounding ($F_1$, %) | | | Argument Grounding ($F_1$, %) | | |
|---|---|---|---|---|---|---|
| Method | Traffic Light | Search and Rescue | Warehouse | Traffic Light | Search and Rescue | Warehouse |
| GPT-3.5 Turbo | 73.5 | 95.0 | 71.1 | 93.9 | 94.0 | 47.7 |
| GPT-4.1 Mini | 76.4 | 87.7 | 98.4 | 94.8 | 90.9 | 51.6 |
| GPT-4o | 85.9 | 94.9 | 82.4 | 87.0 | 95.1 | 70.3 |
| Lang2LTL[†] | - | - | - | 86.2 | 77.6 | 61.8 |
| GinSign (proposed) | 100.0 | 100.0 | 100.0 | **97.9** | **91.1** | **94.2** |

**Predicate Grounding Evaluation** GinSign performs perfectly (100%) in all domains, showing that prefix-enumerated classification on a label space of only 4-7 classes (as shown in 2) is solvable by lightweight BERT model. By contrast, GPT-3.5 Turbo and GPT-4.1 Mini lag substantially, especially in the Warehouse domain, where GPT-3.5 reaches only 71.1%. These results support our hierarchical decomposition: predicates are quite easily isolated, which promises to assist GinSign's generalization through reliable filtering.

**Argument Grounding Evaluation** This task remains the key bottleneck, since each prediction must be drawn from dozens or hundreds of type-compatible constants. Table 2 shows that there are between 44 and 175 classes in the argument-space of each domain, making argument grounding significantly more difficult. GPT-4.1 Mini attains impressive performance in Traffic Light and Search-and-Rescue, but its performance collapses in Warehouse (51.6%). Lang2LTL struggles here as well (61.8% in Warehouse), reflecting the limitations of embedding-similarity when constants are lexically diverse. GinSign, in contrast, maintains robust performance across all domains ($\geq$90%), outperforming both LLM prompting and Lang2LTL by a large margin. We conclude that the reliable filtering information obtained by GinSign's accurate predicate grounding enables notably higher performance on the argument grounding task by virtue of label space reduction.

### 5.3 END-TO-END TRANSLATION EVALUATION

Table 4 evaluates full NL-to-TL translation. We report both **Logical Equivalence (LE)**, which checks syntactic correctness of the temporal operators and lifted APs, while **Grounded Logical Equivalence (GLE)**, further requires that every AP is correctly grounded. As stated in Section 4, GinSign uses the same BERT lifting model and T5 lifted translation model employed in previous work Chen et al. (2023); English et al. (2025a).

**Logical equivalence.** Prior seq2seq frameworks (NL2TL and Lang2LTL) all achieve near-perfect LE scores (95-100%). This suggests that lifting-based pipelines reliably capture operator structure. NL2LTL, which relies purely on prompting, lags at $\approx$42%.

**Grounded logical equivalence.** Here the differences are stark. No prior work except Lang2LTL attempts grounding, so GLE cannot be measured on these frameworks. Lang2LTL, which does

Table 4: **End-to-end Translation** evaluation results.

| Baseline | Traffic Light | | Search and Rescue | | Warehouse | |
|---|---|---|---|---|---|---|
| | LE (%) | GLE (%) | LE (%) | GLE (%) | LE (%) | GLE (%) |
| NL2LTL (GPT-4.1) [†] | 43.6 | 38.4 | 41.8 | 35.4 | 42.6 | 26.2 |
| NL2TL [†] | 98.7 | 60.1 | 95.0 | 54.4 | 99.0 | 46.2 |
| Lang2LTL | 100.0 | 73.6 | 100.0 | 59.0 | 100.0 | 38.8 |
| GinSign (Proposed) | 100.0 | **98.3** | 100.0 | **93.4** | 100.0 | **95.0** |

attempt grounding, suffers a significant reduction in accuracy, achieving GLE (73.6% in Traffic Light, 59.0% in S&R, 38.8% in Warehouse). In contrast, GinSign achieves $\geq$93% in all domains, including 95.0% in Warehouse, representing more than a **2.4**$\times$ absolute gain over the strongest prior method in this domain. By stress testing the grounding components of these two frameworks, we reveal the high cost incurred by inaccurate AP grounding.

### 5.4 GROUNDING ABLATION

Here we ablate the BERT grounding model used for predicate and argument grounding in GinSign. We perform two ablations. First, in Table 5, we evaluate three models: one trained only on predicate grounding, one trained only on argument grounding, and one trained jointly on both tasks. All three models are trained on all domains but with partial domain signatures, and they are tested on the portions of the signatures held out during training. The list of held-out predicates and arguments for each signature is provided in Appendix A.5. Next, in Table 6, we evaluate the same three models, but each is trained on only two of the three domains at a time, using the full domain signatures, to assess how grounding generalizes to unseen domains.

Our results in Table 5 show that grounding maps natural language into elements of a system signature in a way that is not only generalizable by a single model but also improves out-of-distribution performance. Despite having no exposure to the predicates and arguments included in this evaluation during training, the jointly trained model matches or exceeds the performance of the specialized (pred or arg only) models. In Table 6, the diagonal entries show how GinSign generalizes to unseen domains. We find that GinSign consistently achieves high accuracy and respectable $F_1$ scores across all out-of-domain evaluations. Unsurprisingly, under this training regime, GinSign performs better on in-distribution domains than in Table 5, because the latter evaluates only on out-of-distribution data, whereas here the entire signature is in-domain. Overall, these two ablation studies demonstrate the robustness of the proposed GinSign approach, highlighting its ability to generalize to both unseen grounding tasks (predicates or arguments) and unseen domains.

Table 5: Evaluation of Intra-Domain OOD performance on the Predicate and Argument grounding task.

| Model | Task | Traffic Light | | Search & Rescue | | Warehouse | |
|---|---|---|---|---|---|---|---|
| | | Acc | $F_1$ | Acc | $F_1$ | Acc | $F_1$ |
| Pred Only | Predicate | 75.0 | 69.7 | **92.7** | 83.9 | 83.9 | 71.6 |
| Joint | Predicate | **83.1** | **80.3** | **92.7** | **85.5** | **85.1** | **73.0** |
| Arg Only | Argument | 99.1 | 94.7 | 97.0 | 88.4 | 93.2 | 62.6 |
| Joint | Argument | **99.9** | **99.5** | **99.1** | **96.2** | **94.2** | **66.7** |

**Error Analyses**   Here we discuss the most prevalent failure modes that arose during our evaluations. Firstly, we observe the significantly lower performance exhibited by almost all grounding approaches on the Warehouse domain. In the argument grounding evaluation, the mean $F_1$ across all approaches is 63.8%, compared to a mean $F_1$ of over 90% over the Traffic Light and Search and Rescue domains. We found that the diverse natural-language references to the `item` arguments make them difficult to ground, leading to frequent errors. Additionally, the LLM-based approaches often failed to identify correct constants on the signature given in the input. This difficulty motivated our development of the hierarchical filtering approach, described in 4.1.

### 6 CONCLUSION

In this paper, we introduce GinSign, the first end-to-end grounded NL-to-TL framework that anchors every atomic proposition to a system signature. Treating grounding as an open-set, hierarchical span-classification task cleanly separates syntactic translation from semantic anchoring. Experiments on VLTL-Bench show that adding the signature prefix essentially solves both predicate and argument grounding, closing the accuracy gap with larger language models and even outperforms them on visually oriented domains. Crucially, explicit grounding enables model-checking evaluation, exposing

Table 6: Evaluation of Cross-Domain OOD performance on the Predicate and Argument grounding task.

| Holdout Domain | Model | Task | Traffic Light | | Search & Rescue | | Warehouse | |
|---|---|---|---|---|---|---|---|---|
| | | | Acc | $F_1$ | Acc | $F_1$ | Acc | $F_1$ |
| Traffic Light | Pred Only | Predicate | 100.0 | 100.0 | 100.0 | 100.0 | 100.0 | 100.0 |
| | Joint | Predicate | 98.9 | 82.0 | 100.0 | 100.0 | 100.0 | 100.0 |
| | Arg Only | Argument | 95.2 | 66.8 | 100.0 | 100.0 | 100.0 | 99.8 |
| | Joint | Argument | 95.6 | 68.2 | 100.0 | 100.0 | 99.8 | 97.8 |
| Search & Rescue | Pred Only | Predicate | 100.0 | 100.0 | 97.2 | 55.4 | 100.0 | 100.0 |
| | Joint | Predicate | 100.0 | 100.0 | 95.7 | 50.7 | 100.0 | 100.0 |
| | Arg Only | Argument | 100.0 | 100.0 | 96.4 | 72.9 | 100.0 | 99.7 |
| | Joint | Argument | 100.0 | 100.0 | 94.0 | 61.4 | 99.8 | 98.2 |
| Warehouse | Pred Only | Predicate | 100.0 | 100.0 | 100.0 | 100.0 | 99.2 | 86.8 |
| | Joint | Predicate | 100.0 | 100.0 | 100.0 | 100.0 | 99.2 | 87.0 |
| | Arg Only | Argument | 100.0 | 100.0 | 100.0 | 100.0 | 96.9 | 63.6 |
| | Joint | Argument | 100.0 | 100.0 | 100.0 | 99.9 | 97.1 | 65.4 |

semantic errors that remain invisible to purely string-based metrics and pushing NL-to-TL translation from seemingly plausible output toward truly verifiable specifications. We hope this work sparks broader adoption of grounded translation benchmarks and inspires future research on scalable grounded translation for richer temporal logics and larger system vocabularies.

**Limitations and Future Work**   GinSign was tested only on VLTL-Bench, whose signatures may not reflect larger or evolving systems. The framework handles propositional LTL; extending it to metric or first-order variants will require grounding for numbers, time bounds, and quantifiers. Constant-level grounding remains the accuracy bottleneck, especially when names are ambiguous, and the method assumes the signature is fixed at inference. Future work should introduce richer benchmarks, add retrieval- or interaction-based grounding to tackle large constant sets, and develop mechanisms that adapt to signature updates without retraining.

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

# A APPENDIX

## A.1 NOTATION

To improve readability and clarity, we summarize the main symbols and notation used throughout the paper.

**Languages, traces, and temporal logic.**

- $\Sigma$ : finite alphabet of observations (e.g., sets of atomic propositions that hold at a time step).
- $\Sigma^\omega$ : set of infinite traces over $\Sigma$ (infinite sequences of observations).
- $\varphi$ : LTL formulas.
- $\pi \in \mathcal{P}$ : atomic proposition symbol.
- $\mathcal{P}$ : (lifted) atomic proposition vocabulary used in LTL formulas.
- $\bigcirc, \Diamond, \Box, \mathcal{U}$ : "next", "eventually", "always", and "until" temporal operators, respectively.
- $\mathcal{M} = (S, S_0, R, L)$ : Kripke structure modeling the implementation, where $S$ is the set of states, $S_0 \subseteq S$ the initial states, $R \subseteq S \times S$ the transition relation, and $L : S \to 2^{\mathcal{P}}$ the labeling function.
- $[\![\varphi]\!] \subseteq \Sigma^\omega$ : set of traces that satisfy the LTL formula $\varphi$.

**System signatures and grounded atoms.**

- $\mathcal{S} = \langle T, P, C \rangle$ : many-sorted system signature.
- $T$ : finite set of *type* symbols (e.g., `location`, `agent`, `item`).
- $P$ : finite set of *predicate* symbols.
- $C$ : finite set of *constant* symbols.
- $\tau_1, \ldots, \tau_m \in T$ : types in the type signature of a predicate.
- $p \in P$ : predicate symbol with arity $\mathrm{arity}(p)$ and type signature $(\tau_1, \ldots, \tau_m) \in T^m$.
- $c \in C$ : constant symbol; we also refer to constants as *arguments*.
- $\mathrm{type}(c) \in T$ : type of constant $c$.
- $p(c_1, \ldots, c_m)$ : grounded atomic proposition obtained by instantiating $p$ with typed constants $c_i \in C$ of appropriate types.
- $\mathcal{P}_{\mathcal{S}}$ : grounded atomic-proposition vocabulary induced by $\mathcal{S}$,

$$\mathcal{P}_{\mathcal{S}} = \{\, p \mid p \in P \,\} \cup \{\, p(c_1, \ldots, c_m) \mid p \in P,\ c_i \in C \,\}.$$

**NL-to-TL translation and grounding.**

- $s$ : natural-language (NL) specification.
- $\mathbb{S}$ : token sequence of an NL specification.
- $f : s \to \varphi$ : NL-to-TL translation function.
- $\lambda : \mathbb{S} \to \mathbb{Z}_{\geq 0}$ : lifting function that maps each token to 0 (non-AP) or to an AP index.
- $\mathrm{prop}_i$ : $i$-th lifted AP appearing in the lifted NL or LTL (e.g., `prop_1`).
- $k$ : number of lifted AP placeholders in a specification.
- $g_{\mathcal{S}}$ : grounding function

$$g_{\mathcal{S}} : \{\mathrm{prop}_1, \ldots, \mathrm{prop}_k\} \to \mathcal{P}_{\mathcal{S}},$$

which maps each lifted placeholder to a grounded atom in $\mathcal{P}_{\mathcal{S}}$.

**Grounding model and prefix-based classification.**

- $x_{\text{AP}}$ : NL span corresponding to a lifted atomic proposition (local AP context).
- $x_{\text{prefix}}$ : tokenized prefix enumerating candidate symbols from the signature.
- $L = [\ell_1, \ldots, \ell_N]$ : ordered list of candidate labels (predicates or type-filtered constants).
- $N$ : number of candidates in $L$.
- $L_p = \text{enum}(P)$ : enumerated predicate candidate list.
- $L_c^{(r)} = \text{enum}\big(\{c \in C \mid \text{type}(c) = \tau_r\}\big)$ : candidate list for the $r$-th argument of a predicate, filtered by type.
- $m$ : maximum prefix window (shard) size used by the classifier.
- $W_j$ : $j$-th shard (window) of $L$,

$$W_j = [\ell_{(j-1)m+1}, \ldots, \ell_{\min(jm, N)}], \quad j = 1, \ldots, \lceil N/m \rceil.$$

- $h_\theta$ : BERT-based grounding model that maps $(x_{\text{AP}}, W_j)$ to a discrete index $\hat{y} \in \{1, \ldots, |W_j|\}$.
- $\ell^\star$ : gold (correct) label in $L$ for a given training instance.
- $\mathcal{L}_{\text{CE}}(\theta)$ : cross-entropy loss over shard positions,

$$\mathcal{L}_{\text{CE}}(\theta) = -\log p_\theta\big(y = \text{index}(\ell^\star \in W_j) \mid x_{\text{AP}}, W_j\big).$$

- $R$ : number of tournament rounds when $N > m$; scales as $R = O(\log_m N)$ in our analysis.

## A.2 COMPUTATION SCALING COMPARISON

In this section, we compare the cost scaling behavior of GinSign to a generative LLM that grounds by conditioning on an enumerated domain signature in the prompt. Let $N$ denote the number of candidate symbols (predicates or type-filtered constants) that must be considered for a single grounding decision.

A prompt-based generative LLM must serialize all $N$ candidates into a single input sequence. Because transformer self-attention scales quadratically with sequence length, the incremental cost of including $N$ candidates in the prompt is at least

$$O(N^2),$$

ignoring constant context terms. Thus, increasing the size of the domain signature directly incurs a quadratic increase in compute and memory.

By contrast, GinSign uses a BERT encoder that never attends over more than a fixed window of $m$ prefix tokens at a time. When $N > m$, we shard the prefix into windows $W_j$ of size at most $m$ and perform a tournament reduction. In each round, the $N$ candidates are partitioned into $\lceil N/m \rceil$ shards, all of which are processed in a single batched BERT forward pass. The per-round sequence length is therefore bounded by $O(m)$, and the per-round cost is $O(N)$ (with $m$ treated as a constant factor).

Each round reduces the candidate set by a factor of approximately $m$, so the number of rounds is

$$R = O(\log_m N).$$

Consequently, the overall token-level complexity of GinSign for a fixed shard size $m$ is

$$O(N \log_m N) = O(N \log N)$$

that is, *near-linear* in the size of the domain signature and requiring only a logarithmic number of fixed-length BERT passes. In contrast, a generative LLM must process a single, increasingly long prompt whose self-attention cost grows quadratically in $N$, making GinSign substantially more scalable for large signatures.

## A.3 SYSTEM SIGNATURE PREFIX

---

**Algorithm 1** Prefix Generation Algorithm with Optional Type

---

1: **Input:** Signature $\mathcal{S} = \langle T, P, C \rangle$, optional parameter `type`
2: Initialize `prefix` as empty list
3: **if** `type` is not provided **then**
4:    **for** each predicate $p \in P$ **do**
5:       `prefix`.append($p$)
6:    **end for**
7: **else**
8:    **for** each constant $c \in C$ **do**
9:       **if** type($c$) = `type` **then**
10:         `prefix`.append($c$)
11:       **end if**
12:    **end for**
13: **end if**
14: **Output:** `prefix`

---

## A.4 SYSTEM SIGNATURES

In this section, we provide the entire system signatures of each domain in VLTL-Bench (English et al., 2025b).

**Search and Rescue**

**Types:**

- Person: `injured_civilian, injured_hostile, injured_person, injured_rescuer, injured_victim, safe_civilian, safe_hostile, safe_person, safe_rescuer, safe_victim, unsafe_civilian, unsafe_person, unsafe_rescuer, unsafe_victim`

- Hazard: `debris, fire_source, flood, gas_leak, unstable_beam, active_debris, active_fire_source, active_flood, active_gas_leak, active_unstable_beam, inactive_debris, inactive_fire_source, inactive_flood, inactive_gas_leak, inactive_unstable_beam, impending_debris, impending_fire_source, impending_flood, impending_gas_leak, impending_unstable_beam, probable_debris, probable_fire_source, probable_flood, probable_gas_leak, probable_unstable_beam, nearest_debris, nearest_fire_source, nearest_flood, nearest_gas_leak, nearest_unstable_beam`

**Predicates:**

- `avoid(Hazard)`
- `communicate(Person)`
- `deliver_aid(Person)`
- `get_help(Person)`
- `go_home()`
- `photo(Hazard)`
- `record(Hazard)`

**Traffic Light**

**Types:**

- Light: `light_north, light_south, light_east, light_west`
- Color: `red, yellow, green`
- Road: (all enumerated roads, e.g., `east_1st_avenue, east_1st_street, ..., west_10th_street`)
- Vehicle: `vehicle, car, bus, truck, motorcycle, motorbike, bicycle`
- Person: `person, pedestrian, jaywalker, cyclist`

**Predicates:**

- `change(Light, Color)`
- `record(Event)`
- `photo(Person), photo(Vehicle)`
- `get_help(Person)`

**Warehouse**

**Types:**

- Item:  aeroplane, apple, backpack, banana, baseball_bat, baseball_glove, bear, bed, bench, bicycle, bird, boat, book, bottle, bowl, broccoli, bus, cake, car, carrot, cat, cell_phone, chair, clock, cow, cup, dining_table, dog, donut, elephant, fire_hydrant, fork, frisbee, giraffe, hair-drier, handbag, horse, hot_dog, keyboard, kite, knife, laptop, microwave, motorbike, mouse, orange, oven, parking_meter, person, pizza, potted_plant, refrigerator, remote, sandwich, scissors, sheep, sink, skateboard, skis, snowboard, sofa, spoon, sports_ball, stop_sign, suitcase, surfboard, teddy-bear, tennis_racket, tie, toaster, toilet, toothbrush, traffic_light, train, truck, tv_monitor, umbrella, vase, wine_glass, zebra
- Location: shelf, loading_dock

**Predicates:**

- deliver(Item, Location)
- pickup(Item)
- search(Item)
- get_help()
- idle()

A.5 HELD-OUT SYSTEM SIGNATURE ELEMENTS

**Warehouse**

**Predicate Holdouts:**

- get_help()
- deliver(Item, Location)

**Argument Holdouts:**

- loading_dock, apple, banana, bench, bicycle, book, bottle, bowl, bus, car, chair, cup, dog, donut, elephant, fork, frisbee, giraffe, keyboard, kite, knife, motorbike, remote

**Search and Rescue**

**Predicate Holdouts:**

- communicate(Person)
- record(Person/Threat)

**Argument Holdouts:**

- debris, flood, probable_flood, active_flood, gas_leak, injured_victim, injured_rescuer, safe_victim, unsafe_victim, active_fire_source, inactive_fire_source, nearest_flood, probable_debris, unstable_beam, active_gas_leak

**Traffic Light**

**Predicate Holdouts:**

- `photo(Traffic_Target, Road)`

**Argument Holdouts:**

- `pedestrian, motorcycle, collision, cyclist, jaywalker, yellow`
- `north_1st_street, north_2nd_street, ..., north_10th_street`
- `south_1st_street, south_2nd_street, south_3rd_street, ..., south_6th_street`
- `east_1st_avenue, east_2nd_avenue, east_3rd_avenue, east_4th_avenue, east_5th_avenue`
- `west_1st_avenue, west_2nd_avenue, west_3rd_avenue, west_4th_avenue, west_5th_avenue, west_6th_avenue`
- `northeast_1st_street, northeast_2nd_street`
- `northwest_1st_street, northwest_2nd_street`
- `southeast_1st_street, southeast_2nd_street`
- `southwest_1st_street, southwest_2nd_street`
- `north_1st_avenue, north_2nd_avenue`
- `south_1st_avenue, south_2nd_avenue`

## A.6 LLM Grounding Prompt

Scenario Configuration: {each scenario configuration given in Appendix A.4.}
Sentence: {sentence}
Lifted Sentence: {lifted_sentence}

Return a dictionary of the types, predicates, and constants for each prop_n in the lifted sentence.
The dictionary should be in this form:
prop_dict: {
"prop_1": {
"action_canon": *string*,
"args_canon": *list of strings*,
},
"prop_2": {
"action_canon": *string*,
"args_canon": *list of strings*,
}
}
Now, predict:
prop_dict:

### A.7 GINSIGN BERT GROUNDER HYPERPARAMETERS

We use the `bert-base-uncased` checkpoint hosted on HuggingFace (Devlin et al., 2018). We apply the following training protocol and hyperparameters to all (Predicate-only, Argument-only, and Joint) grounding models used in our evaluation:

- Learning Rate: $5e^{-5}$
- Epochs: 3
- Batch Size: 16
- Weight Decay: 0.01
- Early Stopping Threshold: $1e - 6$
- Early Stopping Patience: 3
- Shard size $m = 20$

Our training code is available in the supplementary materials, and will be made publicly available upon acceptance. The shard size $m$ could be optimized experimentally, but we find that $m = 20$ is large enough to capture all predicate prefixes, and requires at most 5 tournaments in the case of the largest constant set (Traffic Light street names, see A.4.

### A.8 LARGE LANGUAGE MODEL DISCLOSURE

During the preparation of this paper, the authors employed large language models (LLMs) as assistive tools for limited tasks including proof-reading, text summarization, and the discovery of related work. All substantive research contributions, analyses, and claims presented in this paper were conceived, developed, and verified by the authors. The authors maintain full ownership and responsibility for the content of the paper, including its technical correctness, originality, and scholarly contributions.

