# OpenReview forum: "GinSign: Grounding Natural Language into System Signatures for Temporal Logic Translation"
_ICLR.cc/2026/Conference — Submitted to ICLR 2026_

### Official Review · Reviewer_KhUo · 2025-10-30

**Soundness:** 3
**Presentation:** 2
**Contribution:** 3
**Rating:** 4
**Confidence:** 3

**Summary:**

The paper proposes an end-to-end grounded "natural language to temporal logic" generative model. The paper points out that existing NLP-to-TL transformation methods and models are not usable in practice due to the lacking connections between the generated atomic propositions are linked to the predicates & constants.

For the grounding process, the paper proposes a hierarchical approach that first does predicate grounding and then argument grounding. Both processes are approached as natural language classification tasks based on BERT model, with an additional filtering step to simplify the problem. For grounding classification model, the paper proposes a non-standard formulation that aims to map a text fragment (given by the lifing model) to a candidate system signature symbol (predicted / typed constant). The classification model is obtained by fine-tuning pre-trained BERT.

The paper reports clear gains over GPT3-5/4.1 and Lang2LTL baselines with an ability to generalize over unseen predicates/constants.

**Strengths:**

The paper points out an important shortcoming of existing methods and proposes a sensible approach.

The experimental results are interesting, (partly) demonstrating the advantages of the method.

**Weaknesses:**

The results are VLTL-Bench only.

Parts of the paper are difficult-to-read without domain expertise. Overall, I believe , the paper could have been written in a more accessible way considering that it is a submission to a primarily a machine learning conference.

While the work itself is valuable, parts of the work is clearly highly domain dependent. Personally, I find it hard to clearly estimate the degree of relevance to ICLR community, although it has interesting aspects in terms of the generalization abilities of the proposed BERT based formulation and the potentially increasing interest to formal languages in machine learning research.

**Questions:**

Section 5.1 claims that other datasets like Navigation / Cleanup World / GLTL are not applicable. There is also the DeepLTL dataset (Hahn et al). Even if it is not possible to evaluate in terms of grounding, is it not possible to evaluate in terms of overal LTL synthesis ability?

---

> ### Author Response · Authors · 2025-11-21
> **Response to KhUo**
>
> We thank the reviewer for their detailed comments. We would like to address a number of points raised by the reviewer in hopes of clarifying our contributions.
>
> >Q1: Section 5.1 claims that other datasets like Navigation / Cleanup World / GLTL are not applicable. There is also the DeepLTL dataset (Hahn et al). Even if it is not possible to evaluate in terms of grounding, is it not possible to evaluate in terms of overall LTL synthesis ability?
>
> As noted by the reviewer, we evaluate on 3 VLTL-Bench domains. This is because we propose a grounding module, and therefore must evaluate on a dataset with grounded predicate labels. To our knowledge, there is no other sufficiently complex dataset that includes these labels. The datasets which do include grounded predicates are highly limited, for example CW and GLTL, which contain only 8 predicates across the entire dataset. Additionally, we omit isolated evaluation of LTL synthesis components because we do not propose novel LTL synthesis methods. Our contribution is purely a grounding approach, and we therefore consider an LTL synthesis evaluation without grounding to be out of scope.
>
> We would also like to emphasize that our proposed input-defined classification space over dynamic labels (via signature prefix) is of independent interest to the ML community, beyond the specific formal-methods setting. Traditionally, BERT is trained to classify tokens into a predetermined set of classes/labels. In our approach, there is only one label which is used to select an element of the prefix provided in the input.

---

### Official Review · Reviewer_YeYH · 2025-11-03

**Soundness:** 3
**Presentation:** 3
**Contribution:** 2
**Rating:** 4
**Confidence:** 4

**Summary:**

This paper studies the problem of translating natural language (NL) specifications into temporal logic (TL) by grounding the resulting atomic propositions into system signatures. The proposed framework, GinSign, introduces a hierarchical translation pipeline that separates (1) the deduction of the logical structure (lifted temporal logic) from (2) the grounding of atomic propositions to a predefined system signature composed of types, predicates, and constants.

The key insight is that grounding NL specifications to semantically meaningful system entities produces more executable and interpretable formal specifications. The two-level grounding approach first predicts the predicate (predicate grounding) and then connects it to the appropriate arguments (argument grounding). A key claim is that by reframing grounding as a structured classification task, GinSign can employ smaller encoder-only models instead of expensive LLMs, thereby improving efficiency while maintaining accuracy.

Experiments on the VLTL-Bench benchmark, which includes three domains (Search and Rescue, Traffic Light, and Warehouse), show that GinSign achieves near-perfect predicate grounding across all domains, outperforming GPT-3.5 Turbo, GPT-4.1 Mini, and prior systems such as Lang2LTL. For argument grounding and logical equivalence evaluation, GinSign also consistently outperforms GPT-based and NL2LTL baselines, achieving ≥ 90% grounded logical equivalence accuracy.

**Strengths:**

- The paper is clearly written, and the authors situate their contribution well within the growing literature on natural language to temporal logic translation.
- The two-level grounding process (first predicates, then arguments) is sound and mirrors how logical forms are constructed compositionally.
- The results show that GinSign substantially improves grounding accuracy and logical equivalence over strong LLM baselines, including GPT-4.1.
- Explicitly grounding TL specifications in system signatures could benefit downstream tasks such as automated verification, planning, or control synthesis.

**Weaknesses:**

- While the hierarchical design is reasonable, the technical novelty feels incremental. The improvement largely comes from providing the model with *more structured context* (the system signature and a lifted template) and recasting the task as classification, rather than introducing new learning or reasoning mechanisms.
- The approach assumes access to fully specified system signatures (types, predicates, and constants). Although the authors acknowledge this limitation, real-world domains often provide only partial or evolving ontologies, which limits the method’s general applicability.
- Because the evaluation requires access to both lifted and grounded atomic propositions, widely used datasets could not be used, which limits cross-domain validation.
- The framework relies heavily on prompt engineering and structured templates rather than principled modeling. The paper does not discuss *why* hierarchical separation or classification-based formulation helps beyond empirical evidence, or what insights generalize to other grounding or reasoning tasks.
- It remains unclear whether GinSign can operate effectively when only natural language input is available (without lifted sentences or system signatures).

**Questions:**

1. How would the approach perform if only NL input were available, without lifted sentences or explicit system signatures?
2. Could the model infer or learn parts of the system signature automatically, rather than relying on manual specification?
3. How sensitive are the results to prompt design or structured templates? Would smaller or less informative prompts substantially degrade performance?
4. The paper attributes most grounding errors to linguistic ambiguity. Could the authors elaborate on whether other factors, such as missing predicates in the system signature or type mismatches during argument binding, also contributed to these failures?

---

> ### Author Response · Authors · 2025-11-21
> **Response to YeYH**
>
> We thank the reviewer for their detailed comments. We would like to address a number of points raised by the reviewer in hopes of clarifying our contributions.
>
> >Q1: How would the approach perform if only NL input were available, without lifted sentences or explicit system signatures?
>
> Explicit system signatures are a requirement of grounding, as they define the space of possible grounding targets. This is not a limit inherent to GinSign, but a limit imposed by the nature of the grounding problem. As such, GinSign expects the signature prefix to enumerate all possible grounding targets that exist within a given system. The number of possible grounding targets is significantly reduced by our hierarchical approach described in Section 4.1.
>
> >Q2: Could the model infer or learn parts of the system signature automatically, rather than relying on manual specification?
>
> As we stated in our answer to Q1, GinSign can not infer new elements of a system signature. However, if the system signature changes for any reason, GinSign is capable of grounding into the new signature without retraining. Results demonstrating this capability are shown in our OOD evaluations in Table 5 and Table 6 of Section 5.4. Please see our answer to Q2 of our response to all reviewers.
>
> >Q3: How sensitive are the results to prompt design or structured templates? Would smaller or less informative prompts substantially degrade performance?
>
> GinSign is not a prompt-based approach. We build GinSign on top of a low-cost encoder-only model (BERT), rather than a generative LLM such as GPT. The only information accepted by GinSign is 1. System signature prefix, and 2. An AP reference which must be grounded in the aforementioned signature. The intuition of our design is to use the input as the classification space, rather than training the model to ground predicates into learned classes or specific labels.
>
> >Q4: The paper attributes most grounding errors to linguistic ambiguity. Could the authors elaborate on whether other factors, such as missing predicates in the system signature or type mismatches during argument binding, also contributed to these failures?
>
> Grounding errors incurred by GinSign typically result from a predicate reference that is semantically similar to multiple canonical signature predicates. An example from VLTL-Bench’s Warehouse domain: the item reference “bag” could refer to one of two canonical items that appear in the signature prefix- “backpack” or “handbag”. This type of ambiguity is difficult for GinSign to resolve.

---

### Official Review · Reviewer_Sd4T · 2025-11-07

**Soundness:** 2
**Presentation:** 3
**Contribution:** 2
**Rating:** 4
**Confidence:** 4

**Summary:**

This paper introduces GinSgin to address the “grounding problem” in Natural Language (NL) to Temporal Logic (TL) translation, where most existing pipelines produce "lifted" TL formulas and are not directly suitable for formal verification. The main contribution is a hierarchical grounding approach that runs after standard lifting and translation. It involves a lightweight BERT model that classifies an NL span into a system-defined predicate and then grounds the NL span into specific constants from a set of filtered “prefix”. Experiments on the VLTL-Bench demonstrate that this approach achieves high accuracy and outperforms zero-shot LLM-based baselines on the GLE metric.

**Strengths:**

- This paper clearly identifies and tackles a practical bottleneck of grounding in the NL-to-TL problem that most prior work overlooks.

- The proposed hierarchical decomposition is a straightforward and effective way to reduce the search space and leverage the type checking to improve overall performance.

**Weaknesses:**

- The entire framework depends on a known, fixed, finite, and static system signature $<T,P,C>$, which is a strong and often unrealistic assumption for open-world or evolving systems, weakening the contribution’s practical impact. The paper focuses only on grounding ambiguity, offloading logical and semantic ambiguity to upstream modules.
- The method’s effectiveness and scalability to large-scale real-world signatures are unproven. The OOD claims are not fully supported by a cross-domain generalization test, but rather by the intra-domain test (Table 5).
- The SOTA-beating claims (Table 3) are based on an unfair comparison. GinSign (fine-tuned) is compared against zero-shot, non-SOTA LLMs (GPT3.5/4.1-Mini) using a flat and plain prompt, rather than the hierarchical one that GinSign benefits from.

**Questions:**

- The comparison to LLMs in Table 3 appears unfair. Could the authors provide results for a stronger baseline that prompts an SOTA LLM (e.g., GPT-4o) using the same hierarchical decomposition that GinSign benefits from or other common prompting techniques (e.g., CoT, Self-reflection)?

- How do the authors expect the prefix-sharding and tournament mechanism to scale to significantly larger signatures, for example, a system with thousands of constants? What are the practical computational limits?

- The OOD experiment is intra-domain. Have the authors attempted a cross-domain generalization test (e.g., training on Traffic Light and testing on Warehouse) to validate the "transferable reasoning" claim?

- Could the authors add definitions for formal methods notation (like $\Sigma^\omega$) to improve accessibility for a broader audience?

---

> ### Author Response · Authors · 2025-11-21
> **Response to Sd4T**
>
> We thank the reviewer for their detailed comments. We would like to address a number of points raised by the reviewer in hopes of clarifying our contributions.
>
> First, we would like to point out that while we do assume that the system signature is known, this assumption is far less strong than the competing assumption that fully accurate groundings are available. This is the assumption made in previous works. We would also like to point out that if the system signature is *not* known, then grounding is not possible for any system. The limitation is not one built into our system, but one built into the problem. Additionally, there seems to be a misunderstanding about the type of signature accepted by our framework. There is no need for input signatures to be fixed or static; our OOD evaluation in table 5 reports performance when signatures are expanded to include more predicates and arguments than were included during training. Second, the hierarchical decomposition of grounding into predicates and arguments is a key aspect of our framework design that we felt would not be fair to build into our LLM competitor.
>
> >Q1: The comparison to LLMs in Table 3 appears unfair. Could the authors provide results for a stronger baseline that prompts an SOTA LLM (e.g., GPT-4o) using the same hierarchical decomposition that GinSign benefits from or other common prompting techniques (e.g., CoT, Self-reflection)?
>
> Please see our answer to Q4 in our response to all reviewers.
>
> >Q2: How do the authors expect the prefix-sharding and tournament mechanism to scale to significantly larger signatures, for example, a system with thousands of constants? What are the practical computational limits?
>
> Please see our answer to Q3 in our response to all reviewers.
>
> >Q3: The OOD experiment is intra-domain. Have the authors attempted a cross-domain generalization test (e.g., training on Traffic Light and testing on Warehouse) to validate the "transferable reasoning" claim?
>
> Please see our answer to Q2 in our response to all reviewers.
>
> >Q4: Could the authors add definitions for formal methods notation to improve accessibility for a broader audience?
>
> We are happy to provide a more extensive description of the notation, included now in Appendix Section 1

---

### Official Review · Reviewer_xzXi · 2025-11-11

**Soundness:** 3
**Presentation:** 3
**Contribution:** 3
**Rating:** 8
**Confidence:** 4

**Summary:**

This paper addresses a critical bottleneck in natural language (NL) to temporal logic (TL) translation: the lack of semantic grounding of atomic propositions (APs) to system-specific definitions, which renders existing TL outputs syntactically valid but practically unusable for system verification.

The proposed solution, GinSign, introduces a framework that treats grounding as a hierarchical classification problem against a formal system signature. The key innovation is a two-step process: first grounding a natural language span to a predicate from the signature, and then grounding its arguments from type-filtered constants. This is operationalized using a novel prefix enumeration technique with a masked language model (like BERT), transforming grounding into a scalable, domain-agnostic classification task.
The authors evaluate GinSign on the VLTL-Bench dataset (Search and Rescue, Traffic Light, Warehouse domains), showing that it achieves 95.5% grounded logical equivalence (GLE) (a 1.4x improvement over state-of-the-art (SOTA) methods like Lang2LTL), 100% predicate grounding accuracy across all domains, and robust argument grounding ($\ge$ 90% $F_1$). Critically, GinSign’s outputs support downstream model checking, a capability missing in prior NL-to-TL frameworks.

**Strengths:**

- Originality: The formulation of grounding as a prefix-based hierarchical classification is distinct from all existing approaches (e.g., end-to-end LLM generation, embedding-similarity) and is a highly creative solution.

- Quality: The evaluation is thorough: it includes isolated grounding (to validate individual components), end-to-end translation (to assess full pipeline performance), and OOD ablation (to test generalization). The use of multiple baselines ensures that GinSign’s advantages are not overstated.

- Clarity: Complex components are explained with algorithms and examples, making the framework reproducible for other researchers.

- Significance: Unlike prior work that focuses on syntactic correctness, GinSign prioritizes semantic grounding—this shifts NL-to-TL from a "theoretical exercise" to a tool that can be integrated into formal verification workflows for autonomous systems.

**Weaknesses:**

- Limitations of Evaluation Domains: The empirical validation is confined to the VLTL-Bench. While it contains three distinct domains, the scale and complexity of the system signatures may not fully represent large-scale, real-world systems (e.g., full autonomous vehicle specifications). Performance and scalability on signatures with orders-of-magnitude more constants remain an open question.

- Limited TL Coverage: GinSign only supports propositional LTL. The paper mentions extending to metric LTL (with time bounds) or first-order LTL (with quantifiers) as future work, but it does not discuss the technical challenges of grounding for these variants.
Constant Grounding Bottleneck in Warehouse: While GinSign’s Warehouse argument grounding (94.2% $F_1$) outperforms baselines, it is lower than in other domains (Traffic Light: 97.9%, Search and Rescue: 91.1%). The authors attribute this to lexically diverse constants but do not provide a detailed error analysis (e.g., which constants are most often misgrounded, why). Such analysis could guide targeted improvements.

- Dynamic Signature Handling: The framework assumes static system signatures at inference time. For systems where signatures evolve (e.g., adding new predicates/constants), GinSign would require retraining or reconfiguring the prefix—no strategy for incremental adaptation is proposed.

**Questions:**

1. (About Constant Grounding Error Analysis) For the Warehouse domain, could you provide specific examples of misgrounded constants and explain why the current framework fails in these cases?
2. (About Scalability to Large Signatures) For systems with thousands of constants (e.g., a warehouse with 1,000 unique items), the current shard-based classification may become inefficient. Have you explored retrieval-augmented methods?
3. (About Dynamic Signature Adaptation) For systems where new predicates/constants are added post-deployment, how would you update GinSign without full retraining? Could the prefix-enumeration mechanism be combined with few-shot learning to adapt to new signature elements?

---

> ### Author Response · Authors · 2025-11-21
> **Response to xzXi**
>
> We thank the reviewer for their positive assessment of our work and for the detailed, constructive feedback. We address the main weaknesses and questions below.
>
> >Q1: The framework assumes static system signatures at inference time. For systems where signatures evolve (e.g., adding new predicates/constants), GinSign would require retraining or reconfiguring the prefix—no strategy for incremental adaptation is proposed.
>
> While GinSign does require that the *current* system signature be known at inference time, but this is a requirement of the grounding problem itself rather than a specific limitation of our framework. Importantly, the signature is not fixed once and for all: because the classification space is defined by the input prefix, one can update the signature (e.g., add/remove predicates or constants) simply by changing the prefix, without modifying the model architecture. Our intra-domain and cross-domain OOD evaluations (Tables 5 and 6) already demonstrate that GinSign can operate on signatures that differ from those seen during training. Please see our answer to Q2 in our response to all reviewers.
>
> >Q2: For the Warehouse domain, could you provide specific examples of mis-grounded constants and explain why the current framework fails in these cases?
>
> In Warehouse, most errors arise when a natural-language span is compatible with multiple, lexically similar constants. For example, “bag” can plausibly refer to both “backpack” and “handbag” in the signature, and phrases like “front area” or “near the entrance” can map to several candidate zones. In such cases, GinSign consistently chooses a plausible but not always correct constant, reflecting genuine ambiguity rather than systematic failure on clear-cut examples.
>
> >Q3: For systems with thousands of constants (e.g., a warehouse with 1,000 unique items), the current shard-based classification may become inefficient. Have you explored retrieval-augmented methods?
>
> We have clarified the computational scaling in Appendix 2: the cost grows with the number of input tokens, and prefix-sharding plus tournament selection yields logarithmic depth in the number of shards, remaining practical even for signatures with thousands of constants. We have not yet implemented retrieval-augmented variants, but agree they are a natural extension. Please see our answer to Q3 in our response to all reviewers.
>
> >Q4: For systems where new predicates/constants are added post-deployment, how would you update GinSign without full retraining? Could the prefix-enumeration mechanism be combined with few-shot learning to adapt to new signature elements?
>
> New constants of existing types can be added simply by appending them to the signature prefix; GinSign will treat them as additional candidates at inference time without retraining. For genuinely new predicates or types, a small amount of few-shot fine-tuning on examples featuring those elements would likely suffice, and we view this as promising future work building on the current framework.
>
> We are grateful for the reviewer’s positive assessment and for the suggestions that helped us strengthen the paper, particularly in terms of error analysis, scalability discussion, and adaptation to evolving systems.

---

### Author Response · Authors · 2025-11-21
**Response to all reviewers**

We thank all reviewers for their valuable feedback. We appreciate the strengths acknowledged by multiple reviewers, including:
- [xzXi, Sd4T, YeYH, KhUo] acknowledge that GinSign addresses a clear bottleneck in current NL-to-LTL translation systems.
- [xzXi, YeYH, KhUo] recognize the sound and effective design of our hierarchical grounding methodology, as well as our strong empirical performance over LLM baselines.
- [xzXi, YeYH] appreciate the quality and clarity of our presentation, the significance of semantically grounded LTL in combination with syntactically correct LTL generation, and the relevance of accurate grounding to downstream applications in planning, verification, and control.

In addition to the qualities mentioned above, some concerns were shared by multiple reviewers, including:

>Q1: [xzXi, Sd4T, YeYH, KhUo] Evaluation limited to VLTL-Bench:

We evaluate on all 3 VLTL-Bench domains. This is because we propose a grounding module, and therefore must evaluate on a dataset with grounded predicate labels. To our knowledge, there is no other sufficiently complex dataset that includes these labels. The datasets that do include grounded predicates are highly limited; for example, CW and GLTL contain only 8 predicates across the entire dataset. Additionally, we omit isolated evaluation of LTL synthesis components because we do not propose novel LTL synthesis methods. Our contribution is purely a grounding approach, and we therefore consider an LTL synthesis evaluation without grounding to be out of scope. We hope the reviewers find this explanation adequate, and welcome the suggestion of additional grounded NL-LTL translation datasets to include in our evaluation, permitted that sufficient time remains in the rebuttal period to do so.

>Q2: [Sd4T, YeYH, xzXi] Handling dynamic domains/signatures:

Several reviewers pointed out that GinSign assumes a fully specified, static system signature at inference time. While it is true that the system signature must be fully known, this is a limitation of the grounding task in general, not a specific limitation of GinSign. Table 5 in the paper shows the intra-domain OOD evaluation in which domain prefixes are changed from those that appear during training, i.e., the handling of dynamic domain signatures. Additionally, at the request of reviewer Sd4T, we provide a new experimental results table showing performance on fully OOD domains. We train variations of the GinSign model on 2 out of 3 of the domains, and report grounding results for both predicates and arguments on all 3 domains. These results are now shown in Table 6 in section 5.4. We observe surprisingly high performance on the holdout domains: in the best case, the GinSign Predicate grounder achieves perfect performance on grounding predicates in the Traffic Light domain, despite never being exposed to this data during training. Additionally, the GinSign Argument grounder achieves accuracy and F1 scores of 96.4% and 72.9% respectively on the Search and Rescue domain. We find these results to be encouraging evidence of GinSign’s ability to generalize the grounding task. Even when trained on only 2 different domains, much of the performance transfers to unseen domains for both predicate and argument grounding.

>Q3: [Sd4T, YeYH, xzXi] Scalability to large signatures is unclear

In order to address this concern, we have added an explanation of how GinSign’s computational cost scales with signature size, demonstrating that as the signature token count grows, the cost of grounding with GinSign remains strictly below the cost of grounding with generative LLMs. This information can now be found in Appendix 2 of the paper. We show that for a signature containing $N$ tokens, the prompt-based generative LLM incurs $O(N^2)$ inference cost due to quadratic self-attention over a single long prompt. In contrast, GinSign shards the signature into $N/m$ fixed-size prefix windows of length $m$ and performs a tournament over candidates, yielding a total cost of $O(m)$ (linear in the number of signature tokens), with $O(log_mN)$ tournament rounds. Thus, as the signature grows, the overall cost of grounding with GInSign remains strictly below that of LLM-based prompting, and scales much more favorably for large system signatures.

>Q4: [Sd4T, YeYH] Comparisons to LLM baselines may be insufficient or unfair.

We appreciate this feedback, and have performed an additional evaluation using GPT-4o (a reasoning model) with Chain-of-Thought prompting. The results of this new experiment can be found in the updated Table 3. We find that GinSign still outperforms the generative LLMs, even when using the stronger model with CoT prompting, in all but one case (the LLM performs 4% better on argument grounding Search and Rescue). We would also like to point out that GinSign outperforms GPT-4o on grounding Warehouse argument by 23.9%, and in predicate grounding on all domains.

---

### Author Response · Authors · 2025-12-02
**Message for AC**

GinSign addresses a critical and practical bottleneck in NL-to-LTL translation: the grounding of atomic propositions into system-defined predicates and constants, which is essential for downstream verification but largely ignored by prior work. All four reviewers acknowledged this as a meaningful contribution, with reviewers praising the hierarchical decomposition approach and strong empirical results (95.5% grounded logical equivalence, representing a 1.4× improvement over SOTA). In response to reviewer feedback, we strengthened our submission by (1) adding cross-domain OOD experiments (Table 6) demonstrating that GinSign generalizes surprisingly well to entirely unseen domains—achieving up to 100% predicate grounding accuracy on held-out domains, (2) including new baselines with GPT-4o and chain-of-thought prompting, which GinSign still outperforms in most settings, and (3) providing a formal scalability analysis (Appendix 2) showing that GinSign's cost scales linearly with signature size versus quadratically for LLM-based prompting. We also clarified that the requirement for a known system signature is inherent to the grounding problem itself, not a limitation specific to our framework. We believe the revised submission adequately addresses the reviewers' concerns while demonstrating GinSign's practical value for enabling executable, verifiable NL-to-LTL translation, and that it is likely that the reviewers would have adjusted their scores had the discussion period continued.

---

### Meta-Review · Area_Chair_7gr2 · 2026-01-07

**Summary:**

The paper studies natural language (NL) to temporal logic (TL) translation, identifies the lack of semantic grounding to system-specific definitions as a key challenge, and proposes a hierarchical grounding method that grounds predicates and arguments separately. Reviewers recognize the novelty and effectiveness of the approach, noting that it addresses a practical bottleneck and achieves strong empirical results.

The main concern across reviews is generalization to real-world scenarios. Reviewers 1, 2, and 3 point out that the method assumes a fully specified and static system signature, which is unrealistic for open or evolving systems. This also raises concerns about the limited evaluation on VLTL-Bench. Reviewers 2 and 3 further note that the set of compared LLM baselines is insufficient.

The authors’ rebuttal provides additional experimental results, including evaluations on held-out predicates, arguments, and domains, as well as results of GPT-4o. However, these additions do not fully address the generalization concerns. In particular, Table 5 shows a 25% drop in predicate-grounding accuracy for the traffic-light domain, indicating a high possibility that the 1/4 held-out predicates cannot be correctly grounded. The classification-based grounding approach is unlikely to generalize to unseen labels during training.

Given the mostly negative scores and the limited impact of the rebuttal on the main concerns, I lean toward rejection.

**Reviewer Concerns:**

Concerns about generalization are not adequately addressed.

Concerns about the insufficient LLM baselines also remain, as GPT-4o is currently not a frontier model.

**Reviewer Scores:**

Reviewers 2, 3, and 4 are likely to maintain their scores, since their main concerns are not resolved.

---

### Decision · Program_Chairs · 2026-01-26

Reject